# An assessment of the quality of antenatal care and pregnancy outcomes in a tertiary hospital in Ghana

Seth Amponsah-Tabi[1]*, Edward T. Dassah[1,2], Gerald O. Asubonteng[1], Frank Ankobea[1], John J. K. Annan[1], Ebenezer Senu[3], Stephen Opoku[3], Ebenezer Opoku[4], Henry S. Opare-Addo[1]

1 Department of Obstetrics and Gynaecology, Komfo Anokye Teaching Hospital, Kumasi, Ghana, 2 School of Public Health, Kwame Nkrumah University of Science and Technology, Kumasi, Ghana, 3 Department of Molecular Medicine, School of Medicine and Dentistry, Kwame Nkrumah University of Science and Technology, Kumasi, Ghana, 4 Public Health Unit, Komfo Anokye Teaching Hospital, Kumasi, Ghana

* sethonto@gmail.com

**Data Availability Statement:** All relevant data are within the paper and its Supporting Information files.

## Abstract

### Background

Antenatal care (ANC) is imperative to decreasing adverse pregnancy outcomes and their related maternal mortality. However, in sub-Saharan Africa, increases in ANC coverage have not correlated well with improved maternal and fetal outcomes suggesting the quality of ANC received could be the missing link. This study assessed ANC quality and its effect on adverse pregnancy outcomes among women who delivered at Komfo Anokye Teaching Hospital.

### Methods

A cross-sectional study was conducted among women who delivered at Komfo Anokye Teaching Hospital within the study period. Women were selected through systematic sampling and interviewed using a pretested structured questionnaire as well as review of their medical records. Data were collected on their sociodemographic and reproductive characteristics, care provided during ANC and delivery outcomes. Categorical variables were compared using the $\chi^2$ test. Factors associated with quality of ANC and adverse pregnancy outcomes were assessed using univariate and multivariate logistic regression to generate crude and adjusted odds ratios (ORs) with 95% confidence intervals (CIs). Statistical analyses were performed using SPSS and GraphPad Prism. *P*-values of < 0.05 were considered statistically significant.

### Results

950 women were recruited into the study with mean age of 30.39±5.57 years. Less than one-tenth (7.6%) of the women received good quality ANC, 63.4% had average quality ANC, and 29.0% received poor quality ANC. Increasing educational level and initiating ANC in the first trimester [aOR 0.2; 95%CI 0.08–0.68; *p*<0.001] increased the odds of receiving

**Funding:** the authors received no specific funding for this work.

**Competing interests:** The authors have declared that no competing interests exist.

good quality ANC while being unemployed decreased the odds of receiving good quality ANC [aOR 0.3; 95% CI 0.12–0.65; p = 0.003]. Receiving poor and average quality of ANC were significantly associated with increased likelihood of developing anaemia during pregnancy, preeclampsia with severe features or delivering a low birth weight baby.

## Conclusion

Most women did not receive good quality ANC. High quality ANC should be provided while the women are encouraged to comply with the recommendations during ANC.

## Introduction

The annual global births is about 139 million and about 800 women die daily from preventable pregnancy related complications [1,2]. Every year, 2.1–3.8 million pregnancies end in stillbirths whilst over 2.9 million infants die in their first month of life [1,2]. According to the World Health Organization (WHO), women in their reproductive age (15–49) are constantly exposed to the complications of pregnancy, labour and puerperium such as pre-eclampsia/ eclampsia, anaemia, hemorrhage, maternal and perinatal deaths [3,4]. Unfortunately, majority of these complication and their related maternal deaths occur in low- and middle-income countries where there are weak healthcare systems. In order to reduce these complications, antenatal care (ANC) was instituted to provide comprehensive care for the pregnant woman. ANC is imperative to achieving the Sustainable Development Goals which aim to decrease maternal mortality rate to less than 70 per 100,000 live births by 2030 [5].

Antenatal care is making contact with a skilled health professional in order to receive medical attention and service to take care of the pregnancy [3]. It is a health service delivery afforded to pregnant women with the sole purpose of achieving a favorable outcome for both mother and baby [5]. During ANC, care is given to pregnant women for prevention, early diagnosis and treatment of medical and obstetric complications during the prenatal period [6]. The WHO recommends a minimum of 8 contacts, one in the first trimester, two in the second trimester and five in the third trimester [3]. This is based on the evidence that the initial minimum of 4 visits led to increased perinatal deaths by 15% and decreased maternal satisfaction [7].

Observational studies have repeatedly indicated lower adverse maternal, fetal and neonatal outcomes among those with adequate antenatal care visits [8]. Morbidity and mortality of pregnant women is significantly reduced among pregnant women who attend ANC compared to those who do not [9]. ANC provides appropriate screening, interventions and treatment throughout pregnancy. It educates women to improve nutritional quality during pregnancy and encourages them to deliver in facilities with skilled birth attendants [9]. A study shows that both maternal and neonatal mortality is reduced when pregnant women gain adequate knowledge at the ANC [10].

ANC has been identified as a hallmark of preventive medicine [5]. Early ANC initiation and regular visits are believed to potentially result in positive maternal and fetal outcomes [11].

An increase in ANC coverage has resulted in a remarkable reduction in maternal and perinatal morbidity and mortality in developed countries [2]. However, in sub-Saharan Africa, despite an increase in the number of women seeking antenatal care and skilled birth delivery,

maternal mortality remains high. Moreover, the ANC coverage has not correlated well with maternal and fetal survival in Sub-Saharan Africa [2,12].

ANC attendance in Komfo Anokye Teaching Hospital (KATH) where the study is undertaken increased from 13,566 in 2013 to 15,902 in 2017 [KATH Annual Report 2017]. Maternal and fetal morbidities and mortalities continue to be at unacceptably high levels despite increased ANC coverage. Maternal mortality ratio in KATH increased from 1,130.51 per 100,000 live births in 2013 to 1,296.10 per 100,000 live births in 2017 [KATH Annual Report 2017]. Perinatal mortality rate around the same period also increased from 103.01 per 1000 live births to 124.50 per 1000 live births in the same institution. The quality of care provided at the facility level and utilization of prenatal interventions by the patient may be the missing link. It is therefore imperative to determine different levels of quality and its influence on pregnancy outcomes. This study assessed the quality of ANC and its effect on adverse pregnancy outcomes among women who delivered at KATH.

## Materials and methods

### Study design

An analytical cross-sectional study was conducted among women who delivered at KATH from July to November 2019. Participants who consented, were sampled and interviewed to obtain both predictor and outcome variables at the same time. Their prenatal records were also reviewed in order to obtain further information. The different levels of quality received were ascertained from direct interview and review of their prenatal records. Pregnancy outcomes of interest included maternal anaemia in late pregnancy (from 36 weeks until delivery), pre-eclampsia with severe features for the mother, then stillbirths and low birth weights for the fetus. Exposure in this study was quality ANC services.

### Profile of study site

The study was conducted at the Obstetrics and Gynecology Directorate of KATH, the second largest hospital in Ghana. The hospital is a tertiary institution and a referral center for most health facilities in the middle and northern parts of the country. It is sited in Kumasi which is the second largest city in Ghana. From the hospital's records unit, yearly ANC attendance ranges between 13,566 and 15,902 for the past 5 years, with total deliveries ranging from 8,438 to 11,188. There are three delivery suites in KATH: the main suite where majority of deliveries take place; and a suite each in the special ward and the high dependency unit. Women with hypertensive disorders in pregnancy are managed in the high dependency unit. Average monthly deliveries at KATH were 900; 80% occurring in the main labour suite, 15% at the special ward and 5% at the high dependency unit. Client who presented to the KATH delivery suites were interviewed and ANC record booklet reviewed to ascertain the quality of services received and the outcomes of their pregnancies.

### Study population and sample size calculation

All women who delivered at KATH labour suites within the study period were eligible for inclusion into the study. Women without ANC records and those who never received ANC were excluded. The study participants included groups that vary in terms of quality of antenatal care services (good, average and poor-quality antenatal care services). The sample size was calculated from the formula; $n = \left( \frac{(Z_{\frac{\alpha}{2}} + Z_\beta)^2 \times 2 \times \bar{P}(1-\bar{P})}{(P_1 - P_2)^2} \right.$, Where n is the required sample size for a group, N = total sample size for different levels (2n), $Z_{\alpha/2}$ is the z-value at 95% confidence

interval (1.96), $Z_2$ represents the power set to detect desired difference between groups (0.80), $P_1$ is the estimated percentage of women who are likely to be recipient of good quality ANC (29%) [2], $p_2$ is the estimated number of women who are likely to receive less quality of care (24%) [2], and $\bar{P}$ is the average of $p_1$ and $p_2$. Sample size calculation were done at 95% confidence interval and 80% power. Substituting these parameters gives a sample size (n) of 840. Assuming a 15% non-response rate, a total of 950 study participants were included in the study.

## Definition and scoring of quality of ANC services

There is no universally accepted or recognized tool for assessing quality services at the ANC. Most studies use the number of prenatal attendances as a proxy in measuring quality. This system however, does not take into consideration the interventions carried out during these visits. A scoring system was developed to categorize quality care into good, average (moderate) and poor-quality care in this study. The scoring system for this study combined number of contacts patient made with the health facility, the timing of ANC initiation and the recommended interventions that were actually carried out during those contacts. Interventions used for assessing quality included the following; blood pressure monitoring: hemoglobin analysis: syphilis screening: hepatitis B screening: HIV screening: ultrasonography scan: Intermittent Prophylactic Treatment (IPT): Iron/folate supplementation: tetanus toxoid vaccination: hookworm prophylaxis: urinalysis: maternal education: sickling screening: blood group and rhesus investigation and stool analysis.

The maximum score from number of contacts, timing of ANC initiation and interventions carried out equals 40 marks. Participants were then categorized into good quality antenatal care (score of 31–40) average (moderate) quality of antenatal care (21–30) and poor quality ANC (score of less than or equal to 20) [2,13,14]. The scoring of variables and the frequency distribution of the scores are presented in S1 File, Table 1.

## Data collection

Information was obtained from clients who delivered at KATH labour suites over the study period. The clients were interviewed directly by trained research assistants in a language that both clients and researchers understood using structured questionnaires. The interview was conducted in English language for study participants who could communicate in English, those who could not were interviewed in Asante Twi. Additional information was obtained from hospital records to complete the questionnaire. Information about their ANC attendance was confirmed from the ANC record cards. The questionnaire used to ascertain quality was based on both national and WHO guidelines for adequate prenatal care. Pre-testing was carried out at Obuasi Government Hospital which is 55km from the study site using 32 participants. Needed adjustment to questionnaires and sampling techniques were carried out before main data collections were done.

## Sampling technique

Systematic sampling was used to select participants for the study over the five-month period with an average of 190 parturient per month. The number of study participants in each delivery suite was estimated in proportion to the average monthly deliveries in the suite. Hence, 760 clients were interviewed at the main labour suite, 142 at the special ward and 48 clients at the high dependency unit. Dividing the sample size by the 190 participants to be interviewed each month, we obtained a sampling interval of 5. Hence every fifth woman who presented for delivery was invited to participate in the study. The first woman to be invited was selected by

**Table 1. Sociodemographic and reproductive characteristics of the study participants.**

| Variable | Frequency (N = 950) | Percentage (%) |
|---|---|---|
| Age [mean ± SD], years] | 30.39±5.57 | |
| Parity [Median (IQR)] | 1.0 (0.0–2.0) | |
| Gravidity [median (IQR)] | 2.0 (2.0–4.0) | |
| **Age group (years)** | | |
| ≤ 20 | 58 | 6.1 |
| 21–25 | 104 | 11.0 |
| 26–30 | 310 | 32.6 |
| 31–35 | 312 | 32.8 |
| ≥36 | 166 | 17.5 |
| **Educational Level** | | |
| No formal education | 68 | 7.2 |
| Primary/JHS | 482 | 50.7 |
| Secondary | 223 | 23.5 |
| Tertiary | 177 | 18.6 |
| **Occupation** | | |
| Civil Servant | 120 | 12.6 |
| Farming | 107 | 11.3 |
| Petty trading | 264 | 27.8 |
| Other (SSM, travel industry) | 329 | 34.6 |
| Unemployed | 130 | 13.7 |
| **Marital status** | | |
| Single | 203 | 21.4 |
| Married | 747 | 78.6 |
| **Gravidity** | | |
| 1–3 | 684 | 72.0 |
| 4–6 | 218 | 23.0 |
| 7–9 | 45 | 4.7 |
| ≥10 | 3 | 0.3 |
| **Parity** | | |
| 0 | 398 | 41.9 |
| 1–2 | 376 | 39.6 |
| 3–4 | 133 | 14.0 |
| >4 | 43 | 4.5 |
| **Booking visit (weeks)** | | |
| 4-13(Early booking) | 541 | 56.9 |
| 14 and above (Late booking) | 409 | 43.1 |

JHS: Junior High School; SSM: Small Scale Mining, IQR: Interquartile range; SD: Standard deviation.

simple random sampling (lottery without replacement) among the first five women presenting for delivery in the month. Women who declined to participate in the study were replaced by the next consecutive woman presenting for delivery.

## Ethical clearance

Approval for this study was granted by the Committee on Human Research, Publications and Ethics (CHRPE) of KATH and Kwame Nkrumah University of Science and Technology. Individual written consent was also obtained from clients before enrolling them as study

participants. For parturient below 18 years, written informed consent and assent were obtained from the parents/guardians and the parturient respectively.

## Statistical analysis

Data were entered and cleaned in Microsoft Excel. All statistical analyses were performed using the GraphPad Prism Version 8.0 and SPSS version 26.0 Software. In computing descriptive statistics, categorical variables were summarized as frequencies and percentages whilst continuous variables were presented as mean and standard deviation or median and interquartile where appropriate. Univariate and multivariate logistic regression was used to assess factors associated with good quality antenatal care to generate and crude and adjusted odds ratios (cORs and aORs) respectively with 95% confidence intervals (CIs). Similarly, the univariate and multivariate logistic regression was used to assess the association between quality of antenatal care and adverse pregnancy outcomes. *P*-values less than 0.05 were considered statistically significant.

## Results

### Sociodemographic and reproductive characteristics of study participants

A total of 950 women were enrolled in the study and were included in the statistical analysis. Majority, about one-third of the participants were within the age category 31–35 years (32.8%), closely followed by those in 26–30 years (32.6%) and a few were below 15 years (0.2%). The mean age of the women was 30.39±5.57 years. Most of the study participants had had basic education (Primary or Junior High School) (50.7%) and less than a fifth (18.6%) had completed tertiary education. Participants' employment status ranged from unemployment to civil servants. Most of the study participants were into petty trading (27.8%), a few participants were into farming (11.3%), were civil servants (12.6%) or were unemployed (13.7%) whilst the majority were doing other businesses (34.6%) such as small-scale mining and the travel industry. Over three-quarters (78.6%) of the women were married. Nearly three-quarters (72%) of the women had been pregnant up to three times, and about 40% had not delivered before or had had delivered up to twice. The median parity and gravidity were 1.0 (0.0–2.0) and 2.0 (2.0–4.0) respectively. With respect to booking visit (gestational age at which the pregnant woman initiated antenatal contacts), most participants booked early and therefore Most women (56.9%) started ANC before 14 weeks' gestation. Table 1 displays the sociodemographic and reproductive characteristics of the study participants.

### Assessing different levels of quality antenatal care

Of the 950 participants who delivered at KATH within the study period, 7.6% had good quality ANC, 63.4% average quality and 29.0% had poor quality ANC (Fig 1).

### Bivariate and multivariate analysis of sociodemographic factors associated with quality of Antenatal care services

On univariate analysis, increasing age, gravidity of 4–6, increasing educational level, being married, and initiating ANC within the first 13 weeks' of gestation were significantly associated with higher odds of receiving good quality ANC. Compared to civil servants (formal sector), women employed in the informal sector or those who were unemployed had decreased odds of receiving quality ANC, Table 2. After adjusting for confounding, gravidity of 4–6 [aOR 2.2; 95% CI 1.36–3.71; *p* = 0.003], increasing educational level and starting ANC within 13 weeks' gestational age [aOR 0.2; 95%CI 0.08–0.68; *p*<0.001] remained as independent

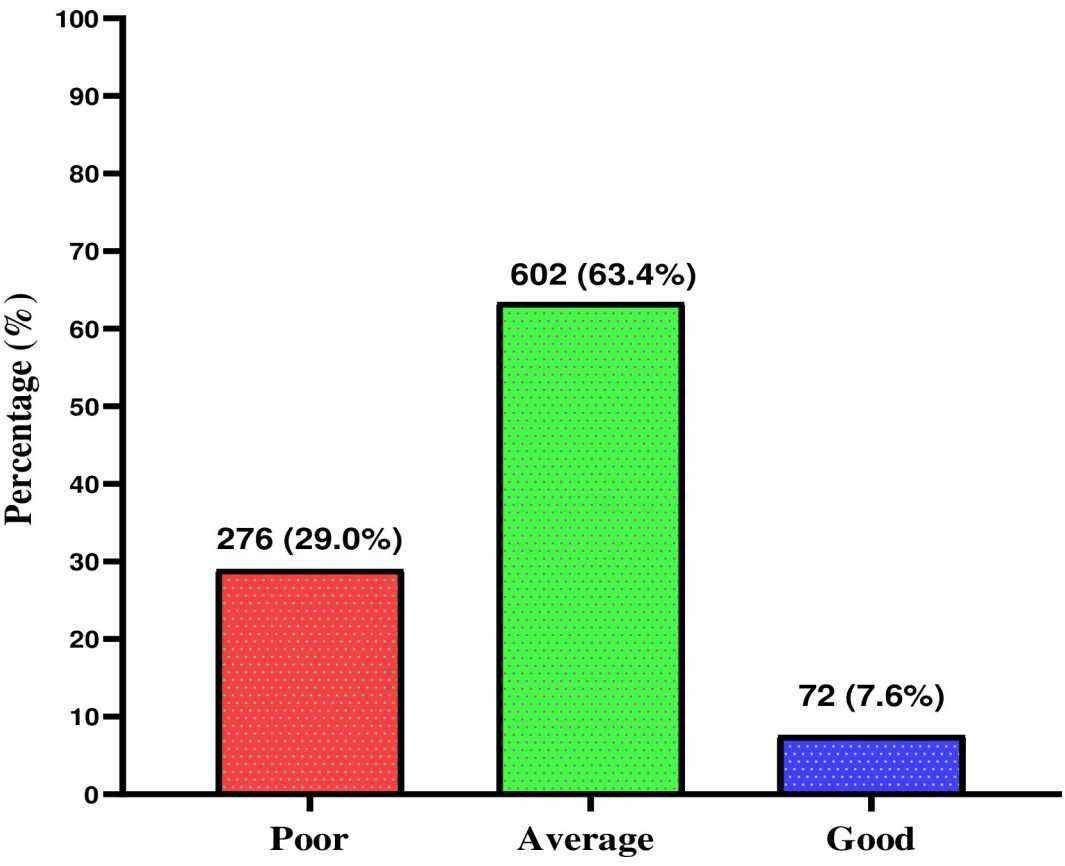

**Fig 1. Distribution of different levels of quality of antenatal care provided to study participants.**

predictors of good quality ANC, Table 2. Women who were unemployed had decreased odds of receiving quality ANC compared to their counterparts who were civil servants [aOR 0.3; 95% CI 0.12–0.65; p = 0.003].

## Adverse pregnancy outcomes

Adverse maternal outcomes were; anaemia in late pregnancy (14.4%), preeclampsia with severe features (31.7%) and eclampsia (5.2%). Preeclampsia with severe features is used to describe a worsening form of hypertensive disorder in pregnancy which includes but not limited to any of the following; systolic blood pressure of 160mmHg or above: diastolic blood pressure of 110mmHg of higher: altered sensorium: persistent headache: visual disturbances: epigastric pain due to involvement of the liver: increased tone in limbs: oliguria: and HELLP syndrome (hemolysis, elevated liver enzymes and low platelets) With respect to the fetal outcomes, 19.1% of the babies were low birth weight and 3.4% were still births, Table 3.

## Quality of ANC and pregnancy outcomes

Women who received poor or average quality ANC were at increased risks of developing anaemia, pre-eclampsia with severe features as well as delivering babies with low birth weight. Compared to women who received good quality ANC, women who received poor quality or average quality ANC had increased odds of developing anaemia in the third trimester, [aOR 10.21; 95% CI 7.89–17.45; $p{<}0.0001$] and [aOR 5.34; 95% CI 3.31–6.76; $p{<}0.0001$]

**Table 2. Sociodemographic and reproductive factors associated with good quality antenatal care.**

| Variable | Good Quality ANC (n = 72) | cOR (95% CI) | *p*-value | aOR (95% CI) | *p*-value |
|---|---|---|---|---|---|
| **Age (years)** | | | | | |
| ≤ 20 | 3 (4.17) | Ref | - | Ref | |
| 21–25 | 13 (18.06) | 3.3 (1.65–6.42) | 0.002 | 1.7 (0.75–3.74) | 0.209 |
| 26–30 | 19 (26.39) | 4.8 (2.65–8.83) | <0.001 | 1.2 (0.56–2.64) | 0.630 |
| 31–35 | 18 (25.0) | 5.2 (2.86–9.54) | <0.001 | 1.1 (0.49–2.54) | 0.787 |
| ≥ 36 | 19 (26.39) | 5.8 (3.05–11.27) | <0.001 | 0.9 (0.44–2.75) | 0.845 |
| **Gravidity** | | | | | |
| 1–3 | 35 (48.61) | Ref | - | Ref | |
| 4–6 | 31 (43.06) | 1.8 (1.24–2.57) | 0.002 | 2.2(1.36–3.71) | **0.003** |
| 7–9 | 5 (6.94) | 1.1 (0.58–2.22) | 0.692 | 2.0(0.88–4.56) | 0.096 |
| ≥10 | 1 (1.39) | 0.9 (0.08–10.30) | 0.952 | 0.8(0.04–13.52) | 0.872 |
| **Parity** | | | | | |
| 0 | 17 (23.61) | Ref | - | - | - |
| 1–2 | 28 (38.89) | 1.3 (0.95–1.76) | 0.107 | - | - |
| 3–4 | 21 (29.17) | 1.4 (0.92–2.24) | 0.112 | - | - |
| >4 | 6 (8.33) | 0.9 (0.50–1.92) | 0.958 | - | - |
| **Educational Level** | | | | | |
| None | 1(1.39) | Ref | - | Ref | - |
| Primary/JHS | 32(44.44) | 1.8 (1.09–3.04) | 0.023 | 2.2 (1.12–4.34) | **0.022** |
| Secondary | 12(16.67) | 1.9 (1.07–3.26) | 0.027 | 2.5 (1.16–5.22) | **0.019** |
| Tertiary | 27(37.50) | 5.3 (2.81–10.16) | <0.001 | 4.7 (1.96–11.49) | **<0.001** |
| **Occupation** | | | | | |
| Civil servant | 22 (30.56) | Ref | - | Ref | - |
| Farmer | 2 (2.78) | 0.5 (0.21–0.85) | 0.015 | 0.8 (0.31–1.97) | 0.596 |
| Petty trading | 21(29.17) | 0.3 (0.18–0.60) | <0.001 | 0.5 (0.24–1.16) | 0.115 |
| Unemployed | 6 (8.33) | 0.1 (0.07–0.24) | <0.001 | 0.3 (0.12–0.65) | **0.003** |
| Others (SMS, travel) | 21 (29.17) | 0.4 (0.23–0.74) | 0.003 | 0.7 (0.33–1.36) | 0.269 |
| **Marital status** | | | | | |
| Single | 10 (13.89) | Ref | - | | - |
| Married | 62 (86.11) | 2.4 (1.74–3.32) | <0.001 | 1.3 (0.86–2.08) | 0.190 |
| **Gestational age of antenatal booking** | | | | | |
| 4-13(early booking) | 66(91.67) | 2.13 (1.93–7.36) | <0.001 | 2.78 (1.51–9.23) | **<0.001** |
| 14-28(late booking) | 6(8.33) | Ref | | Ref | |

ANC: Antenatal Care; CI: Confidence interval, cOR: Crude odds ratio; aOR: Adjusted odds ratio; ANC: Antenatal care; Ref: Reference; JHS: Junior High School.

respectively. Poor quality ANC [aOR 6.35; 95% CI 2.89–9.29; *p*<0.0001] and average quality ANC [aOR 11.21; 95% CI 5.31–22.14; *p*<0.0001] were also associated with increased likelihood of having preeclampsia with severe features. Furthermore, receiving poor quality ANC [aOR 5.43; 95% CI 3.17–8.23; *p*<0.0001] and average quality ANC [aOR 4.34; 95% CI 1.21–5.24; *p*<0.0001] were associated with increased likelihood of having babies with low birth weight, Table 4.

## Discussion

ANC is imperative to decreasing adverse pregnancy outcomes and their related maternal mortality. However, in sub-Saharan Africa, the increase in ANC coverage has not correlated well with improved maternal and fetal outcomes. This suggests that the quality of care provided in

**Table 3. Adverse pregnancy outcomes recorded among study participants.**

| OUTCOMES | Frequency (N = 950) | Percentage (%) |
|---|---|---|
| **Maternal outcomes** | | |
| *Anaemia in late pregnancy* | | |
| Yes | 137 | 14.42 |
| No | 813 | 85.58 |
| *Severe preeclampsia* | | |
| Yes | 301 | 31.68 |
| No | 649 | 68.32 |
| *Eclampsia* | | |
| Yes | 49 | 5.16 |
| No | 901 | 94.84 |
| **Foetal outcomes** | | |
| *Low birth weight* | | |
| Yes | 181 | 19.05 |
| No | 769 | 80.95 |
| *Birth type* | | |
| Live birth | 916 | 96.42 |
| Stillbirth | 34 | 3.58 |

Field study (2019) Author's construct.

healthcare facilities may be the missing link. This study assessed ANC quality and its effect on adverse pregnancy outcomes among women who delivered in a tertiary hospital in Ghana. Less than one-tenth of the women received good quality ANC, nearly two-thirds who had average quality ANC, and almost one-third received poor quality ANC. Independent determinants of quality ANC were increasing educational level, early initiation of ANC and being in formal employment. Poor and average quality of ANC were significantly associated with

**Table 4. Quality of Antenatal care and adverse maternal and fetal outcomes.**

| | ADVERSE MATERNAL OUTCOMES | | | | | |
|---|---|---|---|---|---|---|
| Quality of | Anaemia in pregnancy (n = 137) | | Preeclampsia with severe features (n = 301) | | Eclampsia (n = 49) | |
| ANC | n (%) | aOR (95% CI) | n (%) | aOR (95% CI) | n (%) | aOR (95% CI) |
| Good | 5 (3.6) | Ref (1.00) | 5 (3.6) | Ref (1.00) | 0 (0.0) | Ref (1.00) |
| Average | 24 (17.5) | 5.34 (3.31–6.76)** | 176 (17.5) | 11.21 (5.31–22.14)** | 26 (53.1) | n/a |
| Poor | 108 (78.8) | 10.21(7.89–17.45)*** | 120 (78.8) | 6.35 (2.89–9.29)*** | 23 (46.9) | n/a |
| | FETAL OUTCOMES | | | | | |
| Quality of | Low Birth Weight (n = 181) | | Stillbirth (n = 34) | | | |
| ANC | n (%) | aOR (95% CI) | n (%) | aOR (95% CI) | | |
| Good | 6 (3.3) | Ref (1.00) | 0 (0.0) | Ref (1.00) | | |
| Average | 107 (59.1) | 4.34 (1.21–5.24)*** | 19 (55.9) | n/a | | |
| Poor | 68 (37.6) | 5.43(3.17–8.23)*** | 15 (44.1) | n/a | | |

CI = Confidence interval, aOR = Adjusted odds ratio, ANC = Antenatal care, Ref = Reference, model adjusted for gravidity, gestational age for antenatal booking, occupation and educational level, n/a non-applicable odd ratio due to zero frequency

*$p < 0.05$

**$p < 0.001$

***$p < 0.0001$.

increased likelihood of having anaemia during pregnancy, preeclampsia with severe features and babies with low birth weight.

Our finding of less than one-tenth of the women receiving good quality ANC, while majority had average or poor quality ANC is in consonance with those of a similar study in Zambia where Patricia *et al.* [15] reported a similar finding of low good quality ANC. These findings suggest that although coverage of ANC is almost universal (98%) in Ghana [16],there is the need to improve the quality of care provided to women during pregnancy.

Consistent with the results of previous studies in Ghana [17] and Cambodia [18], we observed that the quality of ANC improved with increasing educational level. This could be attributed to enhanced knowledge about ANC among these groups, which make them more likely to appreciate the benefits of ANC. On the contrary we observed that women who were unemployed or employed in the informal sector and those initiating ANC late were less likely to receive good quality ANC. Women who are not employed or employed in the informal sector may have financial difficulties in accessing health care including ANC which could adversely affect the number of ANC contacts and quality. Similarly, women who initiate ANC late are also likely to have fewer visits and inadequate quality ANC.

The finding of poor or average quality ANC being associated with anaemia during pregnancy is in agreement with those of Wemakor [19] who found poor quality ANC to be associated with anaemia in pregnancy among women receiving ANC at a tertiary referral hospital in Northern Ghana. Interestingly, the prevalence of anaemia in pregnancy in Wemakor's study also increased with the duration of the pregnancy as was observed in this study, with women starting ANC after the first trimester being at increased risk of receiving poor quality ANC compared to their counterparts who booked for ANC in the first trimester. Anaemia in pregnancy could result from several causes including; low dietary iron intake, inadequate or non-compliance with iron supplementation during pregnancy, untreated hookworm and other infestations, as well as undiagnosed or inadequately treated anaemia [20,21]. Malaria which is endemic in the study site is also a known cause of anemia in pregnancy at the sub-region. Intermittent prophylactic treatment has been used as a means to reducing the malaria burden among pregnant women. High quality ANC would ensure women are adequately counselled on nutrition in pregnancy including intake of foods that are rich in iron, women are given adequate and comply with iron supplementation during pregnancy, adequately screened for causes of anaemia in pregnancy, and early diagnosis and treatment of anaemia in pregnancy prior to delivery.

An important finding in this study is that, poor and average quality of ANC were significantly associated with preeclampsia with severe features. This is consistent with Koum *et al.* [22], who found low quality ANC was associated with preeclampsia in Cambodia. This could indicate that primary health care facilities and the human resources needed to provide ANC remain scarce. Prioritizing employee training and effective utilization of limited budgetary and human resources should be employed to expand the coverage of ANC. Training and retraining of health workers and improvement of resources in the health facilities will provide the necessary environment for quality care at the prenatal care.

Again, poor and average quality of ANC were significantly associated with increased likelihood of having babies with low birth weight. Similarly, previous studies from Ghana [23] and other countries [21,22] highlight the association between quality of ANC and LBW. Women who receive high quality ANC including early initiation, adequate number of visits and appropriate interventions during ANC as less likely to have babies with low birth weights. It is imperative that if these women had the required number of antenatal contacts, received care from a skilled provider, and high quality clinical interventions including many of those in our quality assessment score such as health advice and information, and screening and diagnostic

procedures/tests, the incidence of low birth weight babies could have been much lower in this study.

The key strength of our study is that it combined three different quality scoring systems to categorize quality of ANC, thus improving the effectiveness of the system as a tool for assessing the quality of ANC. However, the study has some limitations. First, some indicators of quality of care such as assessing health provider training and client satisfaction with care provided were not included in the scoring system thus limiting the scope of the scoring system. Second, responses to some questions were subject to recall bias due to the retrospective nature of the questions. However, most of these responses were validated with documentations in the maternal record book or hospital records thus minimizing the bias. Finally, the findings from this tertiary referral centre cannot be generalized to the entire population of pregnant women due to selection bias.

## Conclusion

Most pregnant women did not receive good quality ANC. Poor and average quality ANC were associated with adverse maternal and fetal outcomes. Providing high quality ANC would ensure that women are adequately evaluated and treated for anaemia in pregnancy and counselled on prevention of anaemia before delivery. The Ministry of Health and Ghana Health Service should ensure that women receive high quality ANC while the women are encouraged to comply with health education and care provided during ANC. Training and retraining of health workers together with improving health facilities will enable the facility to carry out recommended interventions thus improving quality.

It is imperative to conduct interventional studies to further confirm the association between the quality of ANC and adverse maternal and fetal outcomes.

## Supporting information

**S1 File. This displays ANC quality assessment tool and distribution of interventional scores.**
(DOCX)

**S1 Data.**
(XLS)

## Acknowledgments

### Declarations

The authors are grateful to staff of the Komfo Anokye Teaching Hospital as well as research assistants and volunteers who contributed in diverse ways to the successful implementation of the study.

## Author Contributions

**Conceptualization:** Seth Amponsah-Tabi, Edward T. Dassah, Frank Ankobea, Henry S. Opare-Addo.

**Data curation:** Seth Amponsah-Tabi, Ebenezer Senu, Stephen Opoku.

**Formal analysis:** Seth Amponsah-Tabi, John J. K. Annan, Ebenezer Senu, Stephen Opoku, Ebenezer Opoku.

**Investigation:** Ebenezer Opoku.

**Methodology:** Seth Amponsah-Tabi, John J. K. Annan, Stephen Opoku, Ebenezer Opoku, Henry S. Opare-Addo.

**Project administration:** Edward T. Dassah, Gerald O. Asubonteng, Henry S. Opare-Addo.

**Supervision:** Edward T. Dassah, Gerald O. Asubonteng, Frank Ankobea, John J. K. Annan, Henry S. Opare-Addo.

**Validation:** Gerald O. Asubonteng, Henry S. Opare-Addo.

**Writing – original draft:** Seth Amponsah-Tabi.

**Writing – review & editing:** Seth Amponsah-Tabi.

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
