## [Decision Letter · Decision Letter 0]

15 Jul 2022

PONE-D-22-07633An assessment of the quality of antenatal care and pregnancy outcomes in a tertiary hospital in GhanaPLOS ONE

Dear Dr. Amponsah-Tabi,

Thank you for submitting your manuscript to PLOS ONE. After careful consideration, we feel that it has merit but does not fully meet PLOS ONE’s publication criteria as it currently stands. Therefore, we invite you to submit a revised version of the manuscript that addresses the points raised during the review process.

We look forward to receiving your revised manuscript.

Kind regards,

Muhammad Tarek Abdel Ghafar, M.D

Academic Editor

PLOS ONE

Journal Requirements:

Reviewers' comments:

Reviewer's Responses to Questions

**Comments to the Author**

1. Is the manuscript technically sound, and do the data support the conclusions?

Reviewer #1: Yes

Reviewer #2: Yes

Reviewer #3: Yes

Reviewer #4: Yes

2. Has the statistical analysis been performed appropriately and rigorously? 

Reviewer #1: I Don't Know

Reviewer #2: Yes

Reviewer #3: I Don't Know

Reviewer #4: No

3. Have the authors made all data underlying the findings in their manuscript fully available?

Reviewer #1: Yes

Reviewer #2: Yes

Reviewer #3: Yes

Reviewer #4: Yes

4. Is the manuscript presented in an intelligible fashion and written in standard English?

Reviewer #1: Yes

Reviewer #2: Yes

Reviewer #3: Yes

Reviewer #4: Yes

5. Review Comments to the Author

Reviewer #1: ABSTRACT: Line 33 should read---structured questionnaires as well as a review of their medical records.

METHODS: In line 166, the authors should specify the language/languages used to interview the patients and in line 188, the type of consent obtained from them.

RESULTS: In page 11, table 1, under the variable age -the interval of the ages should be uniform.

In the text the authors should explain what they mean by pre-eclampsia with severe features.

RFERENCES: Numbers 9 and 11 do not seem complete

Reviewer #2: RESULTS: The results were in relevant graphics and tables.

Table 1 (Attachment): it is not clear if this table reflects the total scores or number of participants as it concerns malaria prevention and maternal education; the sum of the figures in these columns are more than 950 ( the total number of participants). This will need to be reviewed as the criteria for scoring are clearly stated and no single person can have more than one score.

DISCUSSION: This section is well written. However, the authors did not emphasize the importance of malaria infection as a cause of anaemia in pregnancy in the tropics. The author in concluding advised the Government to ensure that pregnant women receive high quality ANC through health education and care but did not mention the training and retraining of health workers, and improvement of health facilities

Reviewer #3: This article provides good insight to the poor quality of antenatal care in the given area.

However there needs addition of more details in the methodology. Like what were the recommended interventions during ANC visits? What were the investigations or supplements given? Though it maybe written in the additional file it is better to add main interventions in the methodology.

Also, was the developed tool validated and pretested?

In the results section,

table 4 shows total 31 still births but the total of poor and average ANC care does not cumulate to the same. Please check the numbers.

I would also like to know why were other outcomes not measured like preterm birth, which depends a lot on antenatal care. Knowing the number of preterm babies amongst the LBWs would be beneficial.

Reviewer #4: It seems that this article may reflect the actual circumstance in sub-Saharan Africa. Data collection has been from KATH, second largest hospital in Ghana, also seems high reliability. To prevent the maternal and neonatal serious complications, ANC is well-known to be beneficial. However, the increase of ANC coverage has not correlated to the decrease in number of maternal and fetal death.

This article has written well with adequate discussions. However, some uncertain points remain. Please correct or answer the points as follows;

Minor revision:

In Table 4, the number of stillbirth in average and poor quality of ANC groups suggest 26 and 23, respectively. However, the total number suggested 34. You should re-count or check the actual number.

Major revision:

#1. Despite the number of ANC attendance is around 15,000 yearly, only 950 participants enter the study. The observation period suggested 5 months, so we suppose the major ANC attendance has been ruled out in some reasons. If it is true, this must be the fundamental problem.

#2. The number of participants seems to be full of variety, e.g. age, occupation, education level and gravity. In Table 2, it remains unknown that aORs has actually calculated considering the number of people constitution shown in Table 1. If possible, I would like you to describe the additional comment.

#3. The score of ANC and adverse mother or fetal outcome seems to be correlated. However, the scoring system of ANC seems to remains several concerns to be improved. For example, the Hemoglobin (HB) analysis suggests more than twice with normal result and low HB level with intervention are both 2 points. However, in late preterm period, we often experience that many pregnant women with iron supplementation still remain anemic status. Several other items also should validate the equivalency of points.

6. PLOS authors have the option to publish the peer review history of their article (what does this mean?). If published, this will include your full peer review and any attached files.

Reviewer #1: No

Reviewer #2: No

Reviewer #3: **Yes: **Shreyashi Aryal

Reviewer #4: No

---

## [Author Response · Author response to Decision Letter 0]

4 Sep 2022

REVIEWER 1

#1: ABSTRACT: Line 33 should read---structured questionnaires as well as a review of their medical records.

 Response: Thanks for drawing our attention. Correction carried out in revised manuscript

METHODS: In line 166, the authors should specify the language/languages used to interview the patients and in line 188, the type of consent obtained from them

Response: a. Interview was conducted in English language and Asante Twi (most common local dialect) as depicted in the revised manuscript.

b. Written consent was obtained before enrolling each participant

RESULTS: In page 11, table 1, under the variable age -the interval of the ages should be uniform.

Response: The anomaly is well noted. Necessary changes have been effected in the revised manuscript.

In the text the authors should explain what they mean by pre-eclampsia with severe features.

Response: pre-eclampsia with severe features has been explained in the revised manuscript. 

REFERENCES: Numbers 9 and 11 do not seem complete

Response: Thanks for drawing our attention. The references have been completed in the revised manuscript.

REVIEWER 2 

Q1: Table 1 (Attachment): it is not clear if this table reflects the total scores or number of participants as it concerns malaria prevention and maternal education; the sum of the figures in these columns are more than 950 (the total number of participants). This will need to be reviewed as the criteria for scoring are clearly stated and no single person can have more than one score.

Response: The attached table depicts the proportions or percentages of participants who were recipients of a particular intervention. This was the not the main scoring criteria but gives information on the availability of these interventions as some quality assessment from literature looked at these percentages to connote quality care. 

The figures in the column can be more than the total number of participants. 

Q2: DISCUSSION: This section is well written. However, the authors did not emphasize the importance of malaria infection as a cause of anaemia in pregnancy in the tropics. The author in concluding advised the Government to ensure that pregnant women receive high quality ANC through health education and care but did not mention the training and retraining of health workers, and improvement of health facilities

Response: (a). Thanks for the prompting. Malaria as an important cause of anaemia in pregnancy has been added to the discussion. 

(b). Training and retraining of health workers have been discussed in the revised manuscript.

REVIEWER 3

Q1. Reviewer #3: This article provides good insight to the poor quality of antenatal care in the given area.

However there needs addition of more details in the methodology. Like what were the recommended interventions during ANC visits? What were the investigations or supplements given? Though it may be written in the additional file it is better to add main interventions in the methodology.

 Response: Details of recommended interventions have been added to the methodology.

Q2. Also, was the developed tool validated and pretested?

 Response: The tool was developed from a combination of tools used in Europe and Africa. Pretesting was done.

Q3. In the results section,

table 4 shows total 31 still births but the total of poor and average ANC care does not cumulate to the same. Please check the numbers.

Response: The anomaly has been corrected. Thanks for notification.

Q4. I would also like to know why were other outcomes not measured like preterm birth, which depends a lot on antenatal care. Knowing the number of preterm babies amongst the LBWs would be beneficial

Response: I agree with the submission. The study however did not collect data on those that were preterm or full term. This may be a limitation. Thank you.

REVIEWER 4

MINOR REVISION

In Table 4, the number of stillbirths in average and poor quality of ANC groups suggest 26 and 23, respectively. However, the total number suggested 34. You should re-count or check the actual number

Response: The error in table 4 concerning the number of stillbirths has been noticed and corrected. Thanks for the notification.

MAJOR REVISION

#1. Despite the number of ANC attendance is around 15,000 yearly, only 950 participants enter the study. The observation period suggested 5 months, so we suppose the major ANC attendance has been ruled out in some reasons. If it is true, this must be the fundamental problem

Response: The number of participants enrolled into the study was based on the estimated sample size which has adequate power to achieve the desired effect. During selection, the 950 participants were systematically sampled minimizing selection bias. 

#2. The number of participants seems to be full of variety, e.g. age, occupation, education level and gravity. In Table 2, it remains unknown that aORs has actually calculated considering the number of people constitution shown in Table 1. If possible, I would like you to describe the additional comment

Response: Thanks for the comment. Table 1 shows the sociodemographic and reproductive health factors of all study participants. Table 2 however shows good quality ANC which is the outcome of interest. The odd ratios were calculated based on the number of clients who had good quality care.

#3. The score of ANC and adverse mother or fetal outcome seems to be correlated. However, the scoring system of ANC seems to remains several concerns to be improved. For example, the Hemoglobin (HB) analysis suggests more than twice with normal result and low HB level with intervention are both 2 points. However, in late preterm period, we often experience that many pregnant women with iron supplementation still remain anemic status. Several other items also should validate the equivalency of points.

Response: There is no universal quality scoring system for ANC. This is an attempt to define and score quality based on national and international standards. There may however be portions of this scoring system that can be improved based on further recommendations and studies. Thanks for the notification.

---

## [Decision Letter · Decision Letter 1]

27 Sep 2022

An assessment of the quality of antenatal care and pregnancy outcomes in a tertiary hospital in Ghana

PONE-D-22-07633R1

Dear Dr. Amponsah-Tabi,

We’re pleased to inform you that your manuscript has been judged scientifically suitable for publication and will be formally accepted for publication once it meets all outstanding technical requirements.

Kind regards,

Muhammad Tarek Abdel Ghafar, M.D

Academic Editor

PLOS ONE

Additional Editor Comments (optional):

Reviewers' comments:

Reviewer's Responses to Questions

**Comments to the Author**

1. If the authors have adequately addressed your comments raised in a previous round of review and you feel that this manuscript is now acceptable for publication, you may indicate that here to bypass the “Comments to the Author” section, enter your conflict of interest statement in the “Confidential to Editor” section, and submit your "Accept" recommendation.

Reviewer #1: All comments have been addressed

Reviewer #4: All comments have been addressed

2. Is the manuscript technically sound, and do the data support the conclusions?

Reviewer #1: Yes

Reviewer #4: Yes

3. Has the statistical analysis been performed appropriately and rigorously? 

Reviewer #1: I Don't Know

Reviewer #4: Yes

4. Have the authors made all data underlying the findings in their manuscript fully available?

Reviewer #1: Yes

Reviewer #4: Yes

5. Is the manuscript presented in an intelligible fashion and written in standard English?

Reviewer #1: Yes

Reviewer #4: Yes

6. Review Comments to the Author

Reviewer #1: (No Response)

Reviewer #4: The revised manuscript seems well improved. The response comments to reviewers are also precise and polite.

Thank you for the committed works.

7. PLOS authors have the option to publish the peer review history of their article (what does this mean?). If published, this will include your full peer review and any attached files.

Reviewer #1: No

Reviewer #4: No

---

## [Editor Report · Acceptance letter]

3 Oct 2022

PONE-D-22-07633R1 

An assessment of the quality of antenatal care and pregnancy outcomes in a tertiary hospital in Ghana 

Dear Dr. Amponsah-Tabi:

I'm pleased to inform you that your manuscript has been deemed suitable for publication in PLOS ONE. Congratulations! Your manuscript is now with our production department. 

Kind regards, 

on behalf of

Prof Muhammad Tarek Abdel Ghafar 

Academic Editor

PLOS ONE